# Is a Robust Model More Topologically Smooth? A Comparison of Attention and Convolution

## Abstract

Robustness is a crucial attribute of machine learning models, A robust model ensures consistent performance under input corruptions, adversarial attacks, and out-of-distribution data. While the Wasserstein distance is widely used for assessing robustness by quantifying geometric discrepancies between distributions, its application to layer-wise analysis is limited since computing the Wasserstein distance usually involves dimensionality reduction, which is not suitable for models like CNNs that have layers with diverse output dimensions. To address this, we propose *TopoLip*, a novel metric that facilitates layer-wise robustness analysis. TopoLip enables theoretical and empirical evaluation of robustness, providing insights into how model parameters influence performance. By comparing Transformers and ResNets, we demonstrate that Transformers are more robust in both theoretical settings and experimental evaluations, particularly in handling corrupted and out-of-distribution data.

## 1 Introduction

Robustness is a fundamental aspect of machine learning models (Bai et al., 2021; Wang et al., 2022). Building a robust model has various advantages, which include maintaining high performance under various input corruptions, being resilient to adversarial attack, and generalizing well to out-of-distribution data (Buzhinsky et al., 2023; Szegedy, 2013; Boopathy et al., 2019). When measuring the robustness of models, the Wasserstein distance is always considered (Staerman et al., 2021). The Wasserstein distance measures the geometric discrepancy between probability distributions, making it well-suited for evaluating how models handle shifts or perturbations in input data distributions. Specifically, the Wasserstein distance compares the distributions of inputs and outputs at different stages of a model's processing pipeline, determining how much the distribution of features or predictions changes when input data is altered, either through natural corruptions or adversarial attacks.

When a model processes a dataset and generates outputs, the Wasserstein distance between the input and output distributions can be calculated using optimal transport methods. This typically requires dimensionality reduction to simplify the computation. However, this approach is generally suitable only for evaluating the model as a whole and is less effective for layer-wise analysis. This limitation arises because models like CNNs often have layers with varying output dimensions, and dimensionality reduction in such cases can lead to information loss across different scales.

To circumvent this issue, we propose *TopoLip*, a metric that enables layer-wise analysis. Metrics for robustness are usually abstract and detached from concrete settings (Buzhinsky et al., 2023). By comparing two distinct models: Transformer and ResNet, we demonstrate that the proposed metric is not only robust in experiments but also in concrete theoretical settings (in a specific model). Additionally, TopoLip provides insights into how model parameters influence robustness.

The Transformer architecture, introduced by Vaswani (2017), has become highly popular and has made significant impacts across various fields. In contrast, ResNet, introduced by He et al. (2016), is built using convolutional layers with residual connections. As noted by Bai et al. (2021), Transformers are more robust than CNNs when handling out-of-distribution data. We use these differences in robustness as the basis for our analysis, to guarantee a meaningful result.

This paper is organized as follows: Section 2 presents preliminary concepts; Section 3 introduces the TopoLip robustness metric; Section 4 demonstrates that attention models are more topologically smooth than convolutional models; and Section 5 validates our theoretical findings through experiments. Our main contributions are:

- We propose a new metric for measuring the robustness of models. The metric enables layerwise analysis and concrete theoretical comparison between models. Furthermore, the metric provides insights into the parameter dependency of the model's robustness.

- We propose a relationship between the Lipschitzness of persistence diagrams and the Lipschitzness of probability distributions.

- We investigate the mean-field regime of attention and convolution. By comparing the Wasserstein-Lipschitz condition, we demonstrate that attention layers are more robust to variations in input data distributions.

- We extend the analysis to Vision Transformers (ViTs) and ResNets, demonstrating the same relationship.

- Through experiments, we validate our theoretical findings, demonstrating that attention models are more robust than ResNets when handling corrupted data.

## 1.1 RELATED WORK

**Robustness metric.** Buzhinsky et al. (2023) proposes a metric to measure the robustness of a classifier. This metric is based on probabilistic reasoning within the latent spaces of generative models, which makes it challenging to apply to specific model settings. Similarly, Weng et al. (2018) developed a robustness metric that is attack-independent and can be used with any neural network classifier. However, this approach is not well-suited for the theoretical analysis of individual models.

**Topological Data Analysis** This work builds upon Topological Data Analysis (TDA), which focuses on measuring the topological structures within data. The Wasserstein distance is extensively used in TDA to quantify differences between the topological structures of distributions (Cohen-Steiner et al., 2005). Although persistence diagrams (discussed in Appendix A) are not equivalent to probability spaces, they possess properties that allow for the definition of probability measures (Mileyko et al., 2011). In our study, we further explore the relationship between persistence diagrams and probability spaces, particularly in terms of their Lipschitz continuity.

## 2 PRELIMINARIES

### 2.1 PROBLEM SETUP

Suppose the input is a 2D image with resolution $(H, W)$ and $C$ channels. In Vision Transformers (ViT), the image is reshaped into a sequence of flattened patches $\mathbf{p} \in \mathbb{R}^{N \times (P^2 \cdot C)}$, where $(P, P)$ is the resolution of the patches and $N = \frac{HW}{P^2}$ is the number of patches (Dosovitskiy, 2020). This input is then mapped by an embedding matrix $\mathbf{E} \in \mathbb{R}^{(P^2 \cdot C) \times d}$, where $d$ is the embedding dimension. The mapping yields a matrix of size $\mathbb{R}^{N \times d}$, which can be interpreted as a sequence of $N$ input vectors $\{x_i\}_{i=1}^N \subset \mathbb{R}^d$. These vectors are often expressed as an input matrix $X = [x_1, ..., x_N] \in \mathbb{R}^{d \times N}$.

For a convolutional layer in residual networks (ResNet), let $\mathbf{y}(\alpha) \in \mathbb{R}^C$ represent the input at position $\alpha$. By utilizing a $(2k+1) \times (2k+1)$ filter, the response of a convolutional layer at position $\alpha$ can be written as $\overline{\mathbf{y}}(\alpha) = \sum_{\beta \in ker} W_{:,:,\beta} \phi(\mathbf{y}(\alpha + \beta)) + b$, where $W \in \mathbb{R}^{C \times C \times (2k+1)^2}$ is a weight matrix representing $C$ filters (where we set the $\#(filter) = \#(channel)$), $\phi$ denotes the activation function, and $b \in \mathbb{R}^C$ is a bias term. Since there are $H \times W$ positions at the input image, each corresponding to one response, the input image can be regarded as a $C \times N'$ sequence where $N' = HW$. More details of the convolutional layer setting will be discussed later.

Previous works have restricted the input sequence of the attention layer $X = [x_1, ..., x_N] \in B_R^N$ where $B_R \subset \mathbb{R}^d$ is the closed ball centered at 0 and of radius $R$ (Castin et al., 2024; Geshkovski et al., 2024). We apply this restriction and assume each dimension of $x_i$ $(i \in [N])$ is drawn i.i.d.

from $N(0, \sigma^2)$. Specifically, by applying Chebyshev's inequality that with high probability $1 - d/t^2$, we have $\|x_i\| \leq t\sigma$. For the convolution layer, we assume the input $Y = [y_1, ..., y_{N'}] \in B_R'^{N'}$ where $B_R' \subset \mathbb{R}^C$. Since we set $C$ infinitely large to introduce the mean-field regime of convolution, we instead bound each element: with a high probability $1 - 1/t^2$, we have $|y_{ij}| \leq t\sigma$.

## 2.2 Discrete Frameworks

We define the discrete frameworks of attention and convolution same as the settings in the previous research (He et al., 2015; Chi et al., 2023).

**Definition 1** (Attention layer). *Given an input sequence $X \in \mathbb{R}^{d \times N}$, consider a single-head attention layer with parameters $\{Q_m, K_m, V_m\}_{m \in [M]} \subset \mathbb{R}^{d \times d}$. The output of the single-head attention layer is denoted as $\overline{X} = \text{Attn}_m(X) = [\overline{x}_1, \ldots, \overline{x}_N] \in \mathbb{R}^{d \times N}$, where each $\overline{x}_i$ for $i \in [N]$ is given by*

$$\overline{x}_i = \sum_{j=1}^{N} softmax\left(\frac{x_i^\top Q_m^\top K_m x_j}{\sqrt{d/M}}\right) V_m x_j = \sum_{j=1}^{N} \frac{\exp\left(x_i^\top Q_m^\top K_m x_j / \sqrt{d/M}\right)}{\sum\limits_{k=1}^{N} \exp\left(x_i^\top Q_m^\top K_m x_k / \sqrt{d/M}\right)} V_m x_j.$$

*A multi-head attention extends this concept by allowing the model to attend to information from different representation sub-spaces jointly. A $M$-head attention layer is defined as $\text{MHAttn}(x_i, X) := \boldsymbol{o}_i$, where*

$$\boldsymbol{o}_i = W^O(\oplus_{m=1}^{M} head_m)$$
$$head_m = [\text{Attn}_m(X)]_{:i} = [\text{Attn}(X; \{Q_m, K_m, V_m\})]_{:i},$$

*with $W^O \in \mathbb{R}^{d \times Md}$ being learned projection matrices, and $[A]_{:i}$ denotes the $i$-th column of matrix $A$.*

Next, we define the Transformer with PreLayer Normalization (Pre-LN), which is used in various systems (Fan et al., 2019; Katharopoulos & Fleuret, 2020; Xiong et al., 2019). For a given input vector $x_i \in \mathbb{R}^d$, layer normalization transforms it as $\text{LN}(x_i) = (x_i - \mu_i)/\sigma_i \odot \gamma + \beta$, where $\mu_i = 1/d \sum_{j=1}^{d} x_{i,j}$, $\sigma_i = \sqrt{\sum_{j=1}^{d}(x_{i,j} - \mu_i)^2/d}$, $\gamma \in \mathbb{R}^d$ and $\beta \in \mathbb{R}^d$ are learned scaling and shifting parameters, and $\odot$ denotes element-wise multiplication. An MLP layer with hidden dimension $d'$ is defined as $\text{MLP}(x_i) = W_2\phi(W_1 x_i + b_1) + b_2$ where $W_1 \in \mathbb{R}^{d' \times d}, W_2 \in \mathbb{R}^{d \times d'}, b_1 \in \mathbb{R}^{d'}, b_2 \in \mathbb{R}^d$, and $\phi$ denotes the ReLU function. The Pre-LN Transformer is then expressed as:

$$\text{TF}(X) = \text{MLP} \circ \text{LN}\left(X + \text{MHAttn} \circ \text{LN}(X)\right) + \text{MHAttn} \circ \text{LN}(X) + X.$$

**Definition 2** (Convolutional layer). *Consider a convolutional layer with $C$ filters and $C$ input channels. In practice, each filter could have a different size, and padding is typically applied to maintain consistent output dimensions. To ease the analysis, we set all filters have the same size $(2k + 1) \times (2k + 1)$. Let $y_i(\alpha) \in \mathbb{R}$ represents the input to the convolutional layer with filter $i$ at position $\alpha$, then the output at position $\alpha$ can be written as*

$$\overline{y}_i(\alpha) = \sum_{c=1}^{C} \sum_{\beta \in ker} W_{ci,\beta} \phi(y_c(\alpha + \beta)) + b$$

*where $ker := \{(p_0, p_1) \in \mathbb{Z}^2; |p_0|, |p_1| \leq k\}, W_{ci,\beta} \in \mathbb{R}^{C \times C}$ denotes the weight for from channel $c$ to channel $i$ at position $(\cdot + \beta)$, $b \in \mathbb{R}^C$ is the corresponding bias term, and $\phi$ is the ReLU function.*

Given a mini-batch of size $N$, and a given input sequence of vectors $X = [x_1, ..., x_N] \in \mathbb{R}^{d \times N}$, batch normalization (BN) is applied as $\text{BN}(x_i) = x_i - \mu_B/\sigma_B \odot \gamma + \beta$, where $\mu_B = 1/N \sum_{i=1}^{N} x_i$, $\sigma_B = \sqrt{1/N \sum_{i=1}^{N}(x_i - \mu_B)^2}$. A bottleneck block of ResNet is then expressed as

$$\text{Res}(X) = X + \text{Conv} \circ \text{BN} \circ \text{Conv} \circ \text{BN} \circ \text{Conv} \circ \text{BN}(X).$$

## 2.3 MEAN FIELD FRAMEWORKS

We only define the mean-field attention layer and the mean-field convolution layer here, since our goal is to evaluate the Lipschitzness of models, and the Lipschitzness of the Pre-LN Transformer and the ResNet can be calculated by simply multiplying the Lipschitz numbers of other components.

When the input sequence length $N$ is infinitely large, it can be convenient to model self-attention as a map between probability measures (Sander et al., 2022; Geshkovski et al., 2024; Castin et al., 2024). Indeed, the self-attention map is permutation equivalent, which enables the map from $X = [x_1, ..., x_N]$ to $m(X) = \frac{1}{N} \sum_{i=1}^{N} \delta_{x_i}$.

**Definition 3** (Pushforward (Santambrogio, 2015)). *For a probability measure $\mu$ on $\mathbb{R}^d$ and a measurable map $\varphi : \mathbb{R}^d \to \mathbb{R}^d$, the pushforward of $\mu$ through $\varphi$, denoted as $\varphi_{\#}\mu$, is the probability measure defined by $(\varphi_{\#}\mu)(B) := \mu(\varphi^{-1}(B))$ for any Borel set $B \subset \mathbb{R}^d$, where $\varphi^{-1}(B) := \{x \in \mathbb{R}^d : \varphi(x) \in B\}$.*

**Definition 4** (Mean-field self-attention (Castin et al., 2024)). *Let $Q, K, V \in \mathbb{R}^{d \times d}$, and define $A := K^\top Q / \sqrt{d/M}$. Mean-field self-attention with parameters $(A, V)$ is described as:*

$$F : \mu \in \mathcal{P}_c(\mathbb{R}^d) \mapsto (\Gamma_\mu)_{\#}\mu, \quad \Gamma_\mu(x) = \frac{\int \exp(x^\top A^\top y) V y \, d\mu(y)}{\int \exp(x^\top A^\top y) \, d\mu(y)} \quad for \ x \in \mathbb{R}^d.$$

Since convolution can be permutation equivariant with respect to the channels, it can also be modeled as a map between probability measures. Specifically, the convolutional layer maps the input $Y = [y_1, ..., y_C]$ to $m'(Y) = \frac{1}{C} \sum_{c=1}^{C} \delta_{y_c}$ where $y_i(\alpha) = \sum_\beta W_{c,\beta} \phi(x_c(\alpha + \beta))$ is the response from channel $c$. In previous works, the number of channels is set sufficiently large to make mean field theory applicable (Xiao et al., 2018). Therefore, we can introduce the mean-field convolution based on this limit.

**Definition 5** (Mean-field convolution). *Set $W \in \mathbb{R}^{C \times C \times (2k+1)^2}$. For simplicity, we denote $W_\beta \in \mathbb{R}$ the weight from one channel to another at position $(\cdot + \beta)$. A mean-field convolutional layer with parameter $W$ is described as:*

$$G : \mu' \in \mathcal{P}_c(\mathbb{R}) \mapsto (\Gamma'_{\mu'})_{\#}\mu', \quad \Gamma'_{\mu'}(y(\alpha)) = \int \sum_{\beta \in ker} W_\beta y(\alpha + \beta) d\mu'(Wy) + b$$

*where $ker = \{(\gamma, \eta); (|\gamma|, |\eta| \leq k\}$, $b \in \mathbb{R}$. Here, we ignore the Relu function to ease the analysis.*

## 3 TOPOLOGICAL LIPSCHITZNESS

Before defining Topological Lipschitzness, we first explain the reason that why it is needed.

Wasserstein distance is a notion widely used in optimal transport, defined as a distance function between probability distributions on a given metric space. By considering the Lipschitzness of the Wasserstein distance between the input and the output (which are probability distributions) of a function, instead of considering the Lipschitz continuity between two data points, one can investigate the global behaviors and smoothness of the function (Villani et al., 2009; Villani, 2021). When calculating Wasserstein distance using optimal transport, the computation can be expensive, and dimension reduction is usually employed to facilitate the computation. However, when calculating the Wasserstein distances between layers of a model instead of between the input and output of the whole model, dimension reduction becomes less efficient since models like convolutional neural networks (CNNs) usually include layers with diverse embedding dimensions, causing dimension reduction methods to lose information at different scales.

To address this challenge, we introduce Topological Lipschitzness (TopoLip). TopoLip builds upon Wasserstein Lipschitzness and incorporates concepts from Topological Data Analysis (TDA). A fundamental tool in TDA is persistent homology, which captures multi-scale topological features of data. Persistent homology tracks the evolution of homological structures—such as connected components, loops, and voids—across a nested sequence of spaces $X_1 \subset X_2 \subset \cdots \subset X_n$. Each $k$-dimensional hole in the space $X_i$ is represented in a persistence diagram as a point $(x, y)$, where $x$ and $y$ indicate the scale parameters at which the feature appears (birth) and disappears (death), respectively. Intuitively, this process can be visualized by simultaneously expanding the radius around

each data point: when two expanded points touch, a connection is formed, merging connected components; as the radius continues to grow, higher-dimensional holes may form. For a more detailed explanation, see Appendix A.

Informally, TopoLip measures the Lipschitzness of the Wasserstein distance between the persistence diagrams of a function's input and output. The relationship can be illustrated as follows:

$$\text{Input Distribution} \xrightarrow{\ F\ } \text{Feature Embeddings} \xrightarrow{\ g\ } \text{Persistence Diagrams}$$

$$\downarrow$$

(Probability Distribution)

Here, TopoLip combines the Lipschitzness of the function $F$ with the Lipschitz map $g$ that generates persistence diagrams. Formally, by Lemma 1, TopoLip is defined as belowed:

**Definition 6.** *Let $g$ be a Lipschitz map defined by:*

$$g : \ \mathcal{D} \longrightarrow \mathcal{PD}_k$$

$$g(X) = \{(b_i, d_i)|\ \text{feature } i \text{ in } H_k \text{ births at } b_i \text{ and dies at } d_i\ \}$$

*where $\mathcal{D}$ is the space of finite metric spaces (datasets), and $\mathcal{PD}_k$ is the space of persistence diagrams for dimension $k$ with the Wasserstein distance $W_p$ ($p \geq 1$). For a Lipschitz function $F$, its Topological Lipschitzness is defined as:*

$$\text{Lip}^{W_p}_{\text{TopoLip}}(F) := \text{Lip}^{W_p}(g) \cdot \text{Lip}^{W_p}(F).$$

The map $g$ is Lipschitz due to the stability theorem presented in Cohen-Steiner et al. (2005). When $g$ (in this work, persistent homology) is fixed to generate persistence diagrams, $\text{Lip}(g)$ remains constant. Therefore, the TopoLip of a function is directly proportional to its Wasserstein Lipschitzness. By examining the Wasserstein Lipschitzness of a model, we can gain insights into its TopoLip and overall robustness.

## 4 Wasserstein Lipschitzness comparison

We begin by defining the Wasserstein Lipschitness:

**Definition 7** (Lipschitz constant with respect to the 1-Wasserstein distance (Castin et al., 2024))**.** *Denote $\mathcal{P}_c(\mathbb{R}^d)$ the set of compactly supported probability measures on $\mathbb{R}^d$. P-Wasserstein distance is defined as:*

$$W_p := \left( \inf_{\pi \in \Pi(\mu, \nu)} \int \|x - y\|^p \, d\pi(x, y) \right)^{1/p}$$

*for $\mu, \nu \in \mathcal{P}_c(\mathbb{R}^d)$, where $\Pi(\mu, \nu)$ is the set of couplings between $\mu$ and $\nu$. For a map $F : \mathcal{P}_c(\mathbb{R}^d) \to \mathcal{P}_c(\mathbb{R}^d)$ and any subset $\mathcal{X} \subset \mathcal{P}_c(\mathbb{R}^d)$, the Lipschitz constant of $F$ on $\mathcal{X}$ is defined as:*

$$\text{Lip}^{W_1}(F_\mathcal{X}) := \sup_{\mu, \nu \in \mathcal{X}, \mu \neq \nu} \frac{W_1(F(\mu), F(\nu))}{W_1(\mu, \nu)}.$$

*If $\text{Lip}(F_\mathcal{X})$ is finite, then $F$ is said to be $W_2$-Lipschitz continuous on $\mathcal{P}_c(\mathbb{R}^d)$.*

The reason for using $\text{Lip}^{W_1}$ instead of $\text{Lip}^{W_2}$ here is because for probability measures $\mu$ and $\nu$, $W_1(\mu, \nu) \leq W_2(\mu, \nu)$ holds, meaning that the 1-Wasserstein Lipschitzness can be extended to the 2-Wasserstein Lipschitzness.

To ensure a fair comparison of variances between the self-attention and convolutional layers, we take each element of $Q, K, V, W^O$ in the self-attention layer to be drawn i.i.d. from $\mathcal{N}(0, \sigma^2)$. For the convolution layer, to follow common initialization schemes such as He initialization (He et al., 2015), each element of $W$ is drawn from i.i.d. $\mathcal{N}(0, \sigma^2/(C(2k + 1)^2))$. We assume $H, W, C$ in the input image size $H \times W \times C$ are very large. For the self-attention layer, the input is a sequence with size $d \times N$, where $d$ is the embedding dimension and $N = \frac{HW}{P^2}$. For the convolution layer, the input is a sequence with size $C \times N'$ where $N' = HW$.

## 4.1 ATTENTION AND CONVOLUTION

**Theorem 1.** *Let $Q, K, V \in \mathbb{R}^{d \times d}$. For any $t > \sqrt{d}$ and $s \geq \sigma\sqrt{2\log 2}$, with probability at least $\min\{1 - d/t^2, 1 - 2e^{-s^2/(2\sigma^2)}\}$, and assuming $\|A\|_{op} \geq 2/\sigma^2$, the mean-field single-head attention map $\mathrm{Attn}_{|\mathcal{P}(B_{t\sigma})}$ with parameter $(Q, K, V)$ is $W_1$-Lipschitz continuous on the set $\mathcal{P}(B_{t\sigma})$, and its Lipschitz constant is bounded by*

$$\mathrm{Lip}^{W_1}(\mathrm{Attn}_{|\mathcal{P}(B_{t\sigma})}) = 2t\sigma(2\sigma\sqrt{d} + s)(1 + t\sigma d^{-1/2}(2\sigma\sqrt{d} + s)^2)$$

*Similarly, the Lipschitz constant of mean-field $M$-head attention map $\mathrm{MHAttn}_{|\mathcal{P}(B_{t\sigma})}$ is bounded by*

$$\mathrm{Lip}^{W_1}(\mathrm{MHAttn}_{|\mathcal{P}(B_{t\sigma})}) = 2t\sigma\sqrt{M}(2\sigma\sqrt{d} + s)^2(1 + t\sigma\sqrt{\frac{M}{d}}(2\sigma\sqrt{d} + s)^2).$$

To simplify the upper bounds, assume $t = p\sqrt{d}$, $s = q\sigma$ for constants $p, q > 0$. Under this assumption, the Lipschitz constants of a single-head and multi-head attention layer can be approximated as follows:

$$\mathrm{Lip}^{W_1}(\mathrm{Attn}_{|\mathcal{P}(B_{t\sigma})}) = \mathcal{O}(\sigma^5 d^2), \ \ \mathrm{Lip}^{W_1}(\mathrm{MHAttn}_{|\mathcal{P}(B_{t\sigma})}) = \mathcal{O}(\sigma^6 d^{5/2} M).$$

**Theorem 2.** *Let $W \in \mathbb{R}^{C \times C \times (2k+1)^2}$ where $W_{ci,\beta} \sim N(0, \frac{\sigma^2}{C(2k+1)^2})$ represents the weight from channel $c$ to channel $i$ at position $(\cdot + \beta)$. Denote the output vector of the mean-field convolutional layer as $\overline{\mathbf{y}}(\alpha) = [\overline{y}_1(\alpha), \cdots, \overline{y}_C(\alpha)]$ where $\overline{y}_i(\alpha) = \int_{\mathbb{R}} \left( \sum_\beta W_{ci,\beta} y_i(\alpha + \beta) + b_i \right) d\mu(Wy)$. For any $t > 0$, with probability at least $1 - 1/t^2$, the Lipchitz constant of the mean-field convolution map $\mathrm{Conv}_{|\mathcal{P}(B_{t\sigma})}$ with parameter $W$ is bounded by*

$$\mathrm{Lip}^{W_1}(\mathrm{Conv}_{|\mathcal{P}(B_{t\sigma})}) = (2k+1)\sqrt{t\sigma C\left(1 + \frac{1}{(2k+1)\sqrt{C}}\right)} = \mathcal{O}(k\sqrt{\sigma C}).$$

*where we assume $t$ to be some moderate positive number to simplify the upper bound.*

From the above bounds, we know that the Wasserstein Lipschitzness of attention layers, as well as their TopoLip and robustness, are highly related to the embedding dimension $d$ and the head number $M$. Since $d$ and $M$ are fixed, we can indicate that $\mathrm{Lip}^{W_1}$ of attention layers remains in a certain range. For convolution layers, since their Wasserstein Lipschitzness is related to the channel number $C$ which usually is not fixed in a model, its robustness tends to be lower than attention layers.

Furthermore, if the bound of $\mathrm{Lip}^{W_1}$ is tight enough, it can represent the scale or dynamics $\mathrm{Lip}^{W_1}$. Suppose the bounds in Theorem 1 and 2 are tight, then we can assess the Lipschitz bounds of both models from a practical perspective. In practice, typical parameter values are often set as follows: $\sigma \sim 10^{-2}$, $d \sim 10^2$, $M \sim 10^1$, $k \sim 10^1$, and $C \sim 10^2$. Under this setting, the Lipschitz bound for multi-head attention is on the order of $\mathcal{O}(10^{-6})$, whereas that for convolutional layers is significantly larger, around $\mathcal{O}(10^1)$. To provide a more concrete comparison, consider the following specific parameter settings: $d = 512$, $M = 8$, $\sigma = 0.05$, $k = 3$, and $C = 512$. Under this setting, $\sigma^5 d^2 \approx 0.08$, $\sigma^6 d^{\frac{5}{2}} M \approx 0.74$, while $k\sqrt{\sigma C} \approx 15$. Furthermore, it is important to note that $C$ is not fixed in practice. For instance, the number of channels in ResNet50 are 64→256→512→1024→2048, which leads to a larger Lipschitz bound for convolutional layers. Therefore, convolution is more unstable under this setting, leading to greater TopoLip and lower robustness.

Theorem 1 and 2 indicates that while $\mathrm{Lip}^{W_2}$ of convolution has a bound that is highly unpredictable, $\mathrm{Lip}^{W_2}$ of attention has a fixed bound, and the bound is relatively tight under practical settings. In a real-life scenario, attention and convolution layers are rarely used solely. Instead, they are one part of the models. To conduct a thorough comparison, we extend our investigation to two widely used models: Vision Transformer (ViT) and residual neural network (ResNet).

## 4.2 ViT AND RESNET

We consider the Pre-Layer Normalized Vision Transformers (Pre-LN ViT) and ResNet. Building upon the calculations presented in Theorems 1 and 2, and utilizing Lemma 1, we derive the following

Lipschitz constants:

$$\text{Lip}^{W_1}(\text{TF}) = (\text{Lip}^{W_1}(\text{MLP}) \cdot \text{Lip}^{W_1}(\text{LN}) + 1) \cdot (1 + \text{Lip}^{W_1}(\text{MHAttn}) \cdot \text{Lip}^{W_1}(\text{LN}))$$

$$= (\|W_1\|_{op}\|W_2\|_{op}\|\gamma\|_\infty + 1)(1 + \|\gamma\|_\infty \text{Lip}^{W_1}(\text{MHAttn}))$$

$$= \mathcal{O}\left(\max\left\{1, \sigma^7 d^3 M, \sigma^{10} d^{9/2} M\right\}\right),$$

$$\text{Lip}^{W_1}(\text{Res}) = 1 + \text{Lip}^{W_1}(\text{Conv})^3 \cdot \text{Lip}^{W_1}(\text{BN})^3 = \mathcal{O}\left(\max\left\{1, k^3 \sigma^{5/2} C^3\right\}\right).$$

From these results, we observe that the Lipschitz constants $\text{Lip}^{W_1}$ for both ViTs and ResNets retain and further magnify the parameter dependencies inherent in their respective attention and convolutional layers. Notably, when considering the same settings as discussed in Section 4.1, we find that $\text{Lip}^{W_1}(\text{TF}) = \mathcal{O}(1)$ for ViTs, whereas $\text{Lip}^{W_1}(\text{Res}) = \mathcal{O}(10^4)$ for ResNets. Additionally, since the number of channels $C$ in ResNet can be very large, the Lipschitz constant for ResNet can become significantly higher than that of ViT. As a result, ViTs tend to have a lower TopoLip value, which means they are smoother in terms of their topological properties compared to ResNets. This smoothness suggests that ViTs are less affected by changes or noise in the input, which could make them more stable and robust in their performance.

## 5 EXPERIMENTAL RESULTS

We conduct experiments using the CIFAR-10 and CIFAR-10C dataset to evaluate the relationship between TopoLip and robustness. Specifically, we train ResNet18/50/101 and three ViTs (small, base, and large) for practical settings. We also train two convolution-only models (Conv) and two attention-only models (Attn), each with small and large configurations, to verify the theoretical results for attention and convolution layers.

For Convs, the small configuration uses up to 64 channels across all layers, while the large configuration scales up to 2048 channels in the final layers. For Attn models, the small version features 4 attention heads with an embedding dimension of 128, whereas the large version uses 12 heads with an embedding dimension of 512. All Convs and Attns have 10 layers. Detailed configurations for Attn, Conv, ResNet, and ViT architectures are provided in Table 1.

We train Attns for 100 epochs, Convs for 200 epochs, ResNets for 100 epochs, and ViTs for 200 epochs on CIFAR10 for each model to reach optimal or near-optimal performance levels under the given configurations. ResNet models achieved validation accuracies exceeding 90%, while ViTs range from 77.8% to 87.0% (Figure 7). Attn models, however, showed much lower validation accuracies, with both configurations remaining below 35%, reflecting the limitations of their simplified architectures. In contrast, Conv models performed significantly better, with the small configuration achieving 57.1% and the large configuration reaching 85.7%, despite their simple designs (Figure 8). From the loss curves, we observed that only the training of the large Attn model failed under its simple configuration. This could be attributed to the behavior of attention layers in the early training stages, where they amplify the importance of certain positions or data points. Once a position is deemed important, its attention score increases, reinforcing its significance as training progresses. Without mechanisms like layer normalization to mitigate this effect, the training process can converge prematurely, hindering further weight updates. In fact, the large Attn model's loss failed to record after the first epoch, causing its loss curve in Figure 8 to appear "lost."

Next, we evaluate the TopoLip of the models to understand their robustness. To measure the Wasserstein distance between the persistence diagrams of the input and output at each layer (or each block for ResNets), we first switch the models to evaluation mode to freeze their parameters. Then, we input the test dataset and collect the outputs from all layers. Using these outputs, we compute their persistence diagrams and calculate the Wasserstein distances between adjacent layers. Next, we compute the **absolute change rate**. If the Wasserstein distances of two adjacent layers are $WD_1$ and $WD_2$, the absolute change rate is defined as $|(WD_2 - WD_1)/WD_1|$. The TopoLip of a model is the maximum absolute change rate observed across all layers. While TopoLip provides certain insights into the robustness of models, we propose that rather than focusing solely on TopoLip, analyzing the entire change rate landscape offers a deeper understanding of the model's robustness.

The results of the absolute change rate and cumulative absolute change rate are shown in Figure 1 to 4. Since ResNets and ViTs have different numbers of layers, we interpolate their results to align

Table 1: Model configurations

| Model | Configuration |
|---|---|
| Attn (small) | 4 heads; embedding dimension: 128    (h4 d128) |
| Attn (large) | 12 heads; embedding dimension: 512    (h12 d512) |
| Conv (small) | #(channel): 3→64→64→64→64→64→64→64→64→64 |
| Conv (large) | #(channel): 3→64→64→128→128→256→512→1024→2048→2048 |
| ViT (small) | 6 heads; embedding dimension: 384    (h6 d384) |
| ViT (base) | 12 heads; embedding dimension: 768    (h12 d768) |
| ViT (large) | 16 heads; embedding dimension: 1024    (h16 d1024) |
| ResNet18 | #(channel): 3→64→64(×2)→128(×2)→256(×2)→512(×2) |
| ResNet50 | #(channel): 3→64→64(×3)→128(×4)→256(×6)→512(×3) |
| ResNet101 | #(channel): 3→64→64(×3)→128(×4)→256(×23)→512(×3) |

them on a normalized scale from layer 0 to 1 for consistent comparison. From Figures 1 and 2, we observe that Convs exhibit higher maximum change rates (TopoLip) compared to Attns, indicating that Attns are more robust and topologically smooth than Convs, which aligns with the theoretical results in Section 4.1. From Figure 3, we see that ResNet models have a higher TopoLip than ViTs, with their change rates displaying more turbulent behavior. This is visualized more clearly in Figure 4. Interestingly, the 2-Wasserstein change rate of the ViT (large) model is comparable to that of ResNet101, suggesting they may exhibit similar levels of robustness. The corresponding Wasserstein distances are shown in Figure 9 and 10.

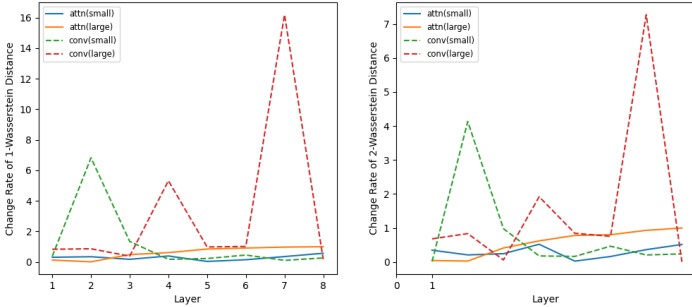

Figure 1: Absolute change rate of the Wasserstein distance of persistence diagrams of Attns and Convs.

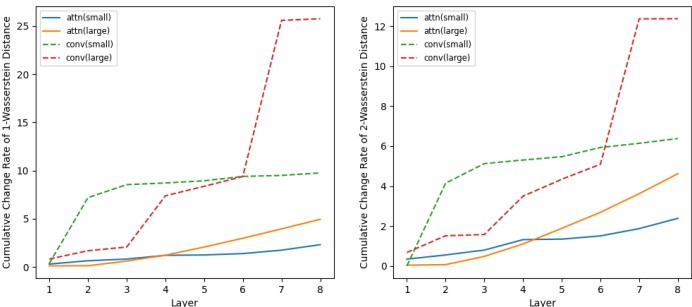

Figure 2: Cumulative absolute change rate of the Wasserstein distance of persistence diagrams of Attns and Convs.

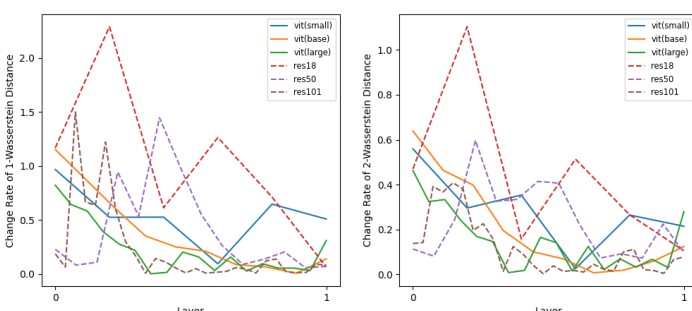

Figure 3: Absolute change rate of the Wasserstein distance of persistence diagrams of ViTs and ResNets.

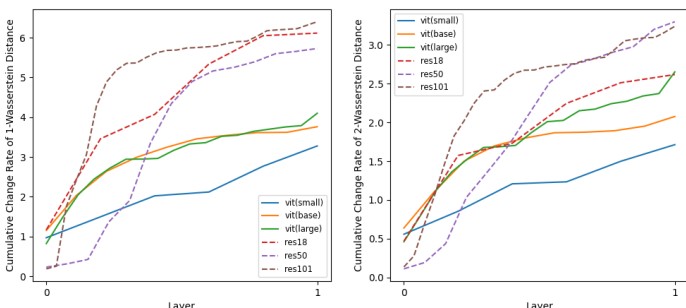

Figure 4: Cumulative absolute change rate of the Wasserstein distance of persistence diagrams of ViTs and ResNets.

Finally, we evaluate the robustness of the models using the CIFAR-10C dataset. CIFAR-10C is an extended version of CIFAR-10, designed to assess model robustness by introducing 15 types of common corruptions, each applied at five severity levels. For our evaluation, we focus on five corruption types: Gaussian Noise, Motion Blur, Snow, Impulse Noise, and Pixelate. The results are summarized in Table 2, where models demonstrating superior robustness (compared to their baselines) are **highlighted**.

From Table 2, we observe that ViTs (small and base) are generally more robust than ResNets across the selected corruption tasks. However, ViT (large) does not exhibit the same level of robustness as its smaller counterparts. Instead, its robustness appears closer to that of ResNets. This behavior is further supported by Figure 3 (right) and 4 (right), where the TopoLip and the dynamics of the 2-Wasserstein change rate curve for ViT (large) are shown to be similar to those of ResNets, potentially explaining this phenomenon.

For Attn and Conv models, while Attn models demonstrate significant robustness, we hypothesize that this is not solely due to the architecture itself. Instead, the limited training capacity of Attn models in such simple configurations likely results in low baseline performance, which can make even slight improvements appear substantial in comparison. Overall, the robustness test results align well with the Wasserstein change rate findings, indicating that TopoLip is closely associated with the robustness of the models.

## 6 CONCLUSION

In this paper, we introduced *TopoLip*, a novel metric for assessing the robustness of machine learning models at the layer-wise level and can provide insights into parameter dependency of models' robustness. We theoretically analyzed the Wasserstein-Lipschitz conditions of the mean-field atten-

Table 2: Model Performance on CIFAR-10 and CIFAR-10C Corruptions (%).

| Model | CIFAR-10 | Gauss Noise | Motion Blur | Snow | Impulse Noise | Pixelate |
|---|---|---|---|---|---|---|
| Attn (small) | 34.2 | **36.5 (+2.3)** | **32.5 (-1.7)** | **30.0 (-4.2)** | **35.2 (+1.0)** | **35.2 (+1.0)** |
| Attn (large) | 12.0 | **16.9 (+4.9)** | **17.2 (+5.2)** | **14.8 (+2.8)** | **16.9 (+4.9)** | **17.0 (+5.0)** |
| Conv (small) | 57.1 | 31.3 (-25.8) | 36.5 (-20.6) | 38.0 (-19.1) | 22.6 (-34.5) | **42.7 (-14.4)** |
| Conv (large) | 85.7 | 44.4 (-41.3) | 51.2 (-34.5) | 61.6 (-24.1) | 23.8 (-61.9) | 60.8 (-24.9) |
| ViT (small) | 77.8 | **53.4 (-24.4)** | **54.0 (-23.8)** | **55.3 (-22.5)** | 39.1 (-38.7) | **62.7 (-15.1)** |
| ViT (base) | 85.2 | **60.8 (-24.4)** | **69.9 (-15.3)** | **75.8 (-9.4)** | 42.2 (-43.0) | **78.3 (-6.9)** |
| ViT (large) | 87.0 | 41.4 (-45.6) | 39.8 (-47.2) | 36.1 (-50.9) | 40.3 (-46.7) | 41.7 (-45.3) |
| ResNet18 | 90.9 | 50.0 (-40.9) | **63.0 (-27.9)** | **73.3 (-17.6)** | 32.8 (-58.1) | 56.1 (-34.8) |
| ResNet50 | 91.4 | 50.0 (-41.4) | 60.3 (-31.1) | **73.1 (-18.3)** | 33.9 (-57.5) | 54.8 (-36.6) |
| ResNet101 | 91.8 | 51.9 (-39.9) | 61.2 (-30.6) | **75.0 (-16.8)** | 35.1 (-56.7) | 66.6 (-25.2) |

tion and convolution, revealing that attention-based models exhibit greater topological smoothness compared to convolutional models. The finding was validated through experiments, demonstrating the superior robustness of Transformers over ResNets when handling corrupted data.

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

## A    PERSISTENCE HOMOLOGY

We provide an intuitive overview of persistence homology, omitting a formal introduction that can be found in (Le & Yamada, 2018; Bubenik et al., 2015; Naitzat et al., 2020). Filtration is a key technique in capturing the topological features of data. Among various types of filtrations, the Čech complex is widely used. The Čech complex constructs a topological structure by forming simplices based on the intersections of balls with a specific radius centered at each data point (Figure 5). As the radius increases, more simplices are added, allowing the complex to capture topological features at different scales.

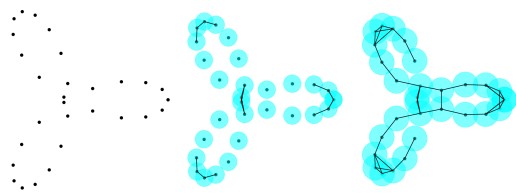

Figure 5: Contruction of the Čech complex of the dataset.

During the filtration process of the Čech complex, topological features such as connected components and holes emerge and disappear. These events are recorded using persistence barcodes, which track the birth and death of each feature (Figure 6). Here, $\beta_0$ and $\beta_1$ represent the lifespan of connected components and 2D holes, respectively. The barcodes are then represented as points in a persistence diagram, which is a multiset of points in the Cartesian plane $\mathbb{R}^2$. In the persistence diagram, $H_0$ corresponds to connected components and $H_1$ to 2D holes. Since the persistence diagram Dg can be considered as a discrete measure $\mu_{Dg} = \sum_{u \in Dg} \delta_u$ where $\delta_u$ is the Dirac unit mass on $u$, the bottleneck distance is usually used to measure the difference between persistence diagrams (Le & Yamada, 2018; Adams et al., 2017). Additionally, 1- and 2-Wasserstein distances are also frequently used (Berwald et al., 2018).

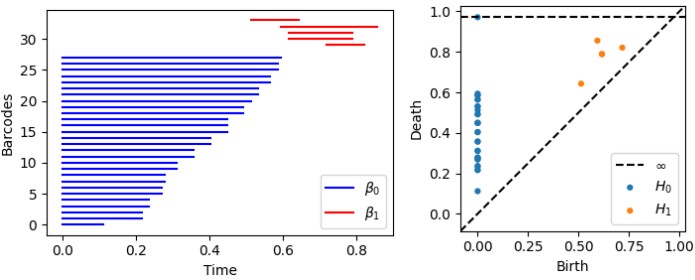

Figure 6: Persistence barcode and persistence diagram.

**Lemma 1** ((Lipschitz Constant of Composed Functions (Gouk et al., 2021))). *Let $(X, d_X)$, $(Y, d_Y)$, and $(Z, d_Z)$ be metric spaces. Suppose that $f : X \to Y$ is Lipschitz continuous with Lipschitz constant $L_f$, and $g : Y \to Z$ is Lipschitz continuous with Lipschitz constant $L_g$. Then the composition $g \circ f : X \to Z$ is Lipschitz continuous with Lipschitz constant at most $L_f \cdot L_g$. In other words, for all $x_1, x_2 \in X$,*

$$d_Z(g(f(x_1)), g(f(x_2))) \leq L_f \cdot L_g \cdot d_X(x_1, x_2).$$

## B    PROOF OF SECTION 3

**Lemma 2** ((Vershynin, 2010)). *Given a matrix $A \in \mathbb{R}^{d \times d}$ with entries $A_{ij} \sim_{i.i.d.} N(0, \sigma^2)$, denote the singular values as $s_1(A) \geq s_2(A) \geq \cdots \geq s_d(A) \geq 0$. Then:*

$$P[s_1 \leq 2\sigma\sqrt{d} + t] \geq 1 - 2e^{-\frac{t^2}{2\sigma^2}}.$$

**Proof of Theorem 1.** We begin by bounding the Lipschitz constant for single-head attention. While Castin et al. (2024) provides an upper bound for $\mathrm{Lip}(\mathrm{Attn}_{|\mathcal{P}(B_{t\sigma})})$, their proof is abbreviated. Here, we present the comprehensive proof and offer a potentially tighter lower bound. We also extend the analysis to multi-head attention by providing an upper bound for $\mathrm{Lip}(\mathrm{MHAttn}_{|\mathcal{P}(B_{t\sigma})})$.

Define the kernel function $K(x,y) := e^{x^\top A^\top y}$. The mean-field attention map is then expressed as:

$$\Gamma_\mu(x) = \int_{\mathbb{R}^d} \frac{K(x,y)Vy}{\int K(x,y)d\mu(y)} d\mu(y).$$

To bound the Lipschitz constant, we consider the difference between $\Gamma_\mu$ and $\Gamma_\nu$ for two probability measures $\mu$ and $\nu$ in $\mathcal{P}(B_{t\sigma})$:

$$\|\Gamma_\mu(x) - \Gamma_\nu(x)\|_{L^\infty(B_{t\sigma}, \mathbb{R}^d)}$$
$$= \left| \frac{\int_{\mathbb{R}^d} K(x,y)Vyd\mu(y)\int_{\mathbb{R}^d}K(x,y)d\nu(y) - \int_{\mathbb{R}^d}K(x,y)Vyd\nu(y)\int_{\mathbb{R}^d}K(x,y)d\mu(y)}{\int_{\mathbb{R}^d}K(x,y)d\mu(y)\int_{\mathbb{R}^d}K(x,y)d\nu(y)} \right|.$$

Denote $y^* := \max_{y \in B_{t\sigma}} \|y\|$. We bound the numerator first:

$$\left| \int_{\mathbb{R}^d} K(x,y)Vyd\mu(y)\int_{\mathbb{R}^d}K(x,y)d\nu(y) - \int_{\mathbb{R}^d}K(x,y)Vyd\nu(y)\int_{\mathbb{R}^d}K(x,y)d\mu(y) \right|$$
$$= \left| \int_{\mathbb{R}^d} K(x,y)Vyd\mu(y)\int_{\mathbb{R}^d}K(x,y)(d\nu - d\mu)(y) \right.$$
$$\left. - \int_{\mathbb{R}^d}K(x,y)Vy(d\nu - d\mu)(y)\int_{\mathbb{R}^d}K(x,y)d\mu(y) \right|$$
$$\leq \left| \int_{\mathbb{R}^d} K(x,y)d\mu(y) \right| \left( \|V\|_{op}y^* \left| \int_{\mathbb{R}^d}K(x,y)(d\nu - d\mu)(y) \right| + \left| \int_{\mathbb{R}^d}K(x,y)Vy(d\nu - d\mu)(y) \right| \right)$$
$$\leq 2\|V\|_{op}y^* \left| \int_{\mathbb{R}^d} K(x,y)d\mu(y) \right| \left| \int_{\mathbb{R}^d}K(x,y)(d\nu - d\mu)(y) \right|$$
$$\leq 2\|V\|_{op}y^* \left| \int_{\mathbb{R}^d} K(x,y)d\mu(y) \right| \|K(x,\cdot)\|_{C^{0,1}(B_{t\sigma})}W_1(\mu,\nu)$$
$$\leq 2y^*\|V\|_{op} \left| \int_{\mathbb{R}^d} K(x,y)d\mu(y) \right| \|K(x,\cdot)\|_{C^{0,1}(B_{t\sigma})}W_2(\mu,\nu)$$

where we use the inequality $W_1(\mu,\nu) \leq W_2(\mu,\nu)$. By Lemma 2, with probability at least $1 - 2e^{-s^2/(2\sigma^2)}$, we have $\|V\|_{op} \leq 2\sigma\sqrt{d} + s$, $\|A\|_{op} \leq \sqrt{\frac{M}{d}}\|K\|_{op}\|Q\|_{op} \leq \sqrt{\frac{M}{d}}(2\sigma\sqrt{d} + s)^2$. For $\|K(x,\cdot)\|_{C^{0,1}(B_{t\sigma})}$, we can bound it as follows:

$$\|K(x,\cdot)\|_{C^{0,1}(B_{t\sigma})}$$
$$= \sup_{y \in B_{t\sigma}} |K(x,y)| + \sup_{y_1 \neq y_2 \in B(0,t\sigma)} \frac{|K(x,y_1) - K(x,y_2)|}{\|y_1 - y_2\|}$$
$$\leq \sup_{y \in B_{t\sigma}} |K(x,y)| + \sup_{y \in B_{t\sigma}} \|\nabla_y K(x,y)\|$$
$$\leq K^*(x,y) + y^*\|A\|_{op}K^*(x,y)$$
$$= K^*(x,y)(1 + y^*\|A\|_{op})$$

where $K^*(x,y) := \sup_{y \in B_{t\sigma}} K(x,y) = \exp(y^*\|x^\top A\|)$ and the first inequality follows from the definition of the $C^{0,1}$ norm and the mean value theorem. Then $\|\Gamma_\mu(x) - \Gamma_\nu(x)\|_{L^\infty(B_{t\sigma}, \mathbb{R}^d)}$ can be

bounded by

$$\|\Gamma_\mu(x) - \Gamma_\nu(x)\|_{L^\infty(B_{t\sigma}, \mathbb{R}^d)}$$

$$\leq \frac{2y^*\|V\|_{op}\left|\int_{\mathbb{R}^d} K(x,y)d\mu(y)\right|K^*(x,y)(1 + y^*\|A\|_{op})}{\left|\int_{\mathbb{R}^d} K(x,y)d\mu(y)\int_{\mathbb{R}^d} K(x,y)d\nu(y)\right|}W_2(\mu,\nu)$$

$$= 2y^*\|V\|_{op}(1 + y^*\|A\|_{op})\frac{K^*(x,y)}{\int_{\mathbb{R}^d} K(x,y)d\nu(y)}W_2(\mu,\nu).$$

To bound the integral part, we transform $\int d\nu(y)$ to $\int p(y)dy$ where $p(y)$ is the probability density function (pdf) of $y$. Since $y \sim N(0, \sigma^2 I)$, by using the pdf of the multivariate Gaussian distribution, we have

$$\int_{R^d} K(x,y)d\nu(y) = \int_{R^d} K(x,y)p(y)dy$$

$$= \frac{1}{(2\pi\sigma^2)^{d/2}}\int_{R^d} e^{x^\top A y} \cdot e^{-\|y\|^2/(2\sigma^2)}dy$$

$$= e^{\sigma^2\|x^\top A\|^2/2}\frac{1}{(2\pi\sigma^2)^{d/2}}\int_{R^d} e^{-\|y - \sigma^2 x^\top A\|^2/(2\sigma^2)}dy$$

$$= e^{\sigma^2\|x^\top A\|^2/2}.$$

Therefore,

$$\frac{K^*(x,y)}{\int_{\mathbb{R}^d} K(x,y)d\nu(y)} = \exp(y^*\|x^\top A\| - \sigma^2\|x^\top A\|^2/2).$$

To bound it at 1, we need to ensure that

$$y^* \leq \frac{\sigma^2}{2}\|x^\top A\| \leq \frac{y^*\sigma^2}{2}\|A\|_{op} \implies \|A\|_{op} \geq \frac{2}{\sigma^2}.$$

holds. Under this condition, the final bound is

$$\|\Gamma_\mu(x) - \Gamma_\nu(x)\|_{L^\infty(B_{t\sigma}, \mathbb{R}^d)} \leq 2y^*\|V\|_{op}(1 + y^*\|A\|_{op})W_2(\mu,\nu) =: \text{Lip}(\text{Attn})W_2(\mu,\nu).$$

Finally, since

$$\Gamma_\mu^{\text{MHAttn}}(x) - \Gamma_\nu^{\text{MHAttn}}(x) = W^O\begin{bmatrix}\Gamma_\mu^1(x) - \Gamma_\nu^1(x) \\ \vdots \\ \Gamma_\mu^M(x) - \Gamma_\nu^M(x)\end{bmatrix}$$

where $\Gamma_\nu^k(x)$ denotes the mean-field self-attention of $k$-th head, we have

$$\|\Gamma_\mu^{\text{MHAttn}}(x) - \Gamma_\nu^{\text{MHAttn}}(x)\|_{L^\infty(B_{t\sigma}, \mathbb{R}^d)}$$

$$\leq \|W^O\|_{op}\left\|\begin{bmatrix}\Gamma_\mu^1(x) - \Gamma_\nu^1(x) \\ \vdots \\ \Gamma_\mu^M(x) - \Gamma_\nu^M(x)\end{bmatrix}\right\|$$

$$\leq \|W^O\|_{op}\sqrt{\sum_{i=1}^M \text{Lip}(\text{Attn}_{|\mathcal{P}(B_{t\sigma})})^2}$$

$$\leq 2y^*\sqrt{M}\|W^O\|_{op}\|V\|_{op}(1 + y^*\|A\|_{op})W_2(\mu,\nu) =: \text{Lip}(\text{MHAttn})W_2(\mu,\nu).$$

With probability at least $\min\{1 - d/t^2, 1 - 2\exp(-s^2/(2\sigma^2))\}$, we can bound the terms by $y^* = t\sigma$, $\|W^O\|_{op}, \|V\|_{op} \leq 2\sigma\sqrt{d} + s$, $\|A\|_{op} \leq \sqrt{M/d}\|K\|_{op}\|Q\|_{op} \leq \sqrt{M/d}(2\sigma\sqrt{d} + s)^2$. Therefore, the final bounds become

$$\|\Gamma_\mu(x) - \Gamma_\nu(x)\|_{L^\infty(B_{t\sigma}, \mathbb{R}^d)} \leq 2t\sigma(2\sigma\sqrt{d} + s)(1 + t\sigma d^{-1/2}(2\sigma\sqrt{d} + s)^2)W_2(\mu,\nu),$$

$$\|\Gamma_\mu^{\text{MHAttn}}(x) - \Gamma_\nu^{\text{MHAttn}}(x)\|_{L^\infty(B_{t\sigma}, \mathbb{R}^d)} \leq 2t\sigma\sqrt{M}(2\sigma\sqrt{d} + s)^2(1 + t\sigma\sqrt{\frac{M}{d}}(2\sigma\sqrt{d} + s)^2)W_2(\mu,\nu)$$

where $M = 1$ for the single-head attention. $\square$

**Proof of Theorem 2.** We begin by bounding the Lipschitz constant for a single response $\overline{y}(\alpha)$. We denote $\overline{y}^\mu(\alpha) = \int_\mathbb{R} \left( \sum_\beta W_\beta y_i(\alpha + \beta) + b_i \right) d\mu(Wy)$, then

$$
\begin{aligned}
& |\overline{y}^\mu(\alpha) - \overline{y}^\nu(\alpha)| \\
&= \left| \int_\mathbb{R} \left( \sum_\beta W_\beta y(\alpha + \beta) + b_i \right) d\mu(Wy) - \int_\mathbb{R} \left( \sum_\beta W_\beta y(\alpha + \beta) + b_i \right) d\nu(Wy) \right| \\
&= \left| \int_\mathbb{R} \left( \sum_\beta W_\beta y(\alpha + \beta) + b_i \right) (d\mu - d\nu)(Wy) \right| \\
&\leq \left\| \left( \nabla_W \left( \sum_\beta W_\beta y(\alpha + \beta) + b_i \right), \nabla_y \left( \sum_\beta W_\beta y(\alpha + \beta) + b_i \right) \right) \right\|_2 W_1(\mu, \nu) \\
&\leq \sqrt{ \left| \sum_\beta y(\alpha + \beta) \right| + \left| \sum_\beta W_\beta \right| } \; W_1(\mu, \nu) \\
&\leq \sqrt{ \sum_\beta |y(\alpha + \beta)| + \sum_\beta |W_\beta| } \; W_1(\mu, \nu) \\
&\leq (2k + 1) \sqrt{ t\sigma + \frac{t\sigma}{(2k+1)\sqrt{C}} } \; W_2(\mu, \nu) =: L\, W_2(\mu, \nu).
\end{aligned}
$$

Finally, since $\Gamma'_\mu(\alpha) = \overline{\mathbf{y}}(\alpha)$, we can bound the difference between $\Gamma'_\mu$ and $\Gamma'_\nu$ as:

$$
\begin{aligned}
\|\Gamma'_\mu(\alpha) - \Gamma'_\nu(x)\|_{L^\infty(B_{t\sigma}, \mathbb{R}^d)} &= \sqrt{ \sum_{i=1}^{C} |\overline{y}^\mu(\alpha) - \overline{y}^\nu(\alpha)|^2 } \\
&\leq \sqrt{C}\, L\, W_2(\mu, \nu) \\
&= (2k+1) \sqrt{ t\sigma C \left( 1 + \frac{1}{(2k+1)\sqrt{C}} \right) } \; W_2(\mu, \nu).
\end{aligned}
$$

$\square$

## C  FURTHER EXPERIMENTAL RESULTS

Figure 7 and 8 demonstrate the training/validation accuracy and loss of models. Figure 9 and 10 demonstrate the Wasserstein distance of the persistence diagrams between adjacent layers of models.

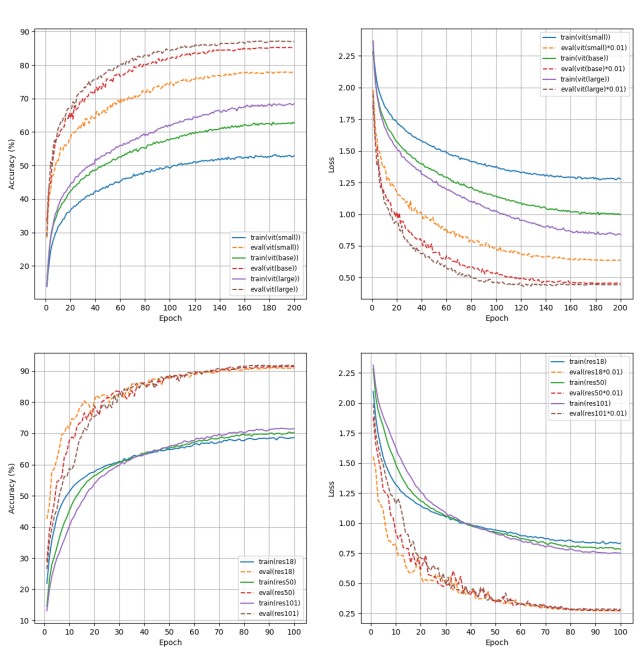

Figure 7: Accuracy and loss of ViTs and ResNets.

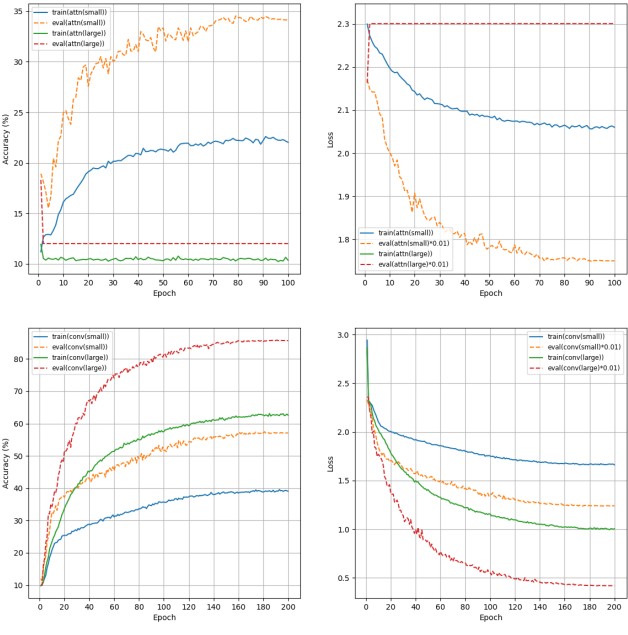

Figure 8: Accuracy and loss of Attns and Convs.

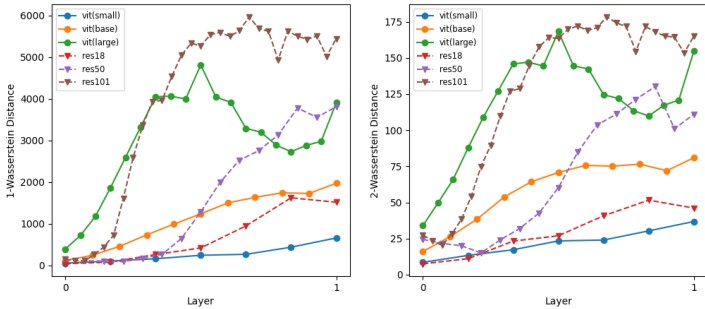

Figure 9: Wasserstein distance of the persistence diagrams of ViTs and ResNets.

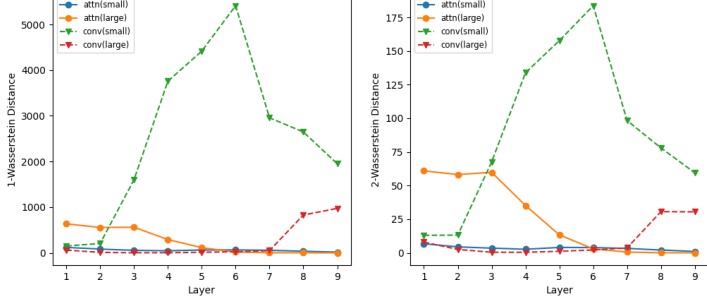

Figure 10: Wasserstein distance of the persistence diagrams of Attns and Convs.

