# OpenReview forum: "Feature Learning in Attention Mechanisms Is More Compact and Stable Than in Convolution"
_ICLR.cc/2025/Conference — ICLR 2025 Conference Withdrawn Submission_

### Official Review · Reviewer_b6Pf · 2024-11-01

**Soundness:** 2
**Presentation:** 1
**Contribution:** 2
**Rating:** 3
**Confidence:** 4

**Summary:**

The paper compares variance, stability, and intrinsic dimensionality of the representations learned via attention or convolution in transformers and CNNs. The authors first study this in a theoretical setting imposing assumptions on the input and formulation of the models, deriving the variance and Lipschitz constants for each of attention and convolution. They then provide arguments about intrinsic dimensionality and how they believe different components of the models (e.g., MLP or batch norm) impact these properties. In the end, they provide experimental observations on variance and stability of two attention and two convolution models as well as ViT and ResNet.

**Strengths:**

Understanding various aspects of success of commonly-used and successful models such as CNNs and transformers is important. This paper focuses on the stability of these models, which could shed light on another factor that plays a role in the remarkable success of transformers. Theoretical results in a clear setup are helpful, despite (some inevitable) shortcomings.

Some examples of interesting and insightful observations/findings provided in the paper:
- Dependency of variance of the outputs in the attention on dimensionality, while this variance only depends on the input variance in convolution (line 285).
- The implications of Theorem 4, showing that the Lipschitz constant of the Attn with practically-relevant values of the variables ends up being much smaller than that of Conv (line 346).
- Potentially intriguing arguments provided in 3.4 for intrinsic dimensionality,
- The experiments are ample for a theoretical paper (but not if the authors believe their main results come from the experiments) and the experimental setup seems suitable to investigate the main questions posed by the paper.

**Weaknesses:**

**The key weakness:** The main motivation of the paper seems lost. The paper does not motivate why a comparison on the variance of stability of transformers and CNNs is a question worth a paper, especially given the limited scope of the setup. Unless the introduction motivates this question, the paper essentially seems like it could be a section of a more thorough study, e.g., on the stability of transformers or on the comparison between CNNs and transformers. While it is not impossible that I am missing something, I do not see a way that this paper could be prepared for publication at ICLR without a major reconsideration of the content and the presentation.

**Other major weaknesses:**

1. Related to the key weakness above: the second paragraph of the introduction seems to jump from some general statements about transformers and CNNs (first paragraph) to questions the authors pose, without explaining why the questions are raised or why their answers matter. Same issue applies to the contributions listed in the introduction.

2. The literature review seems insufficient and shallow. The literature reviewed in the Related Work section is both old, and not closely related to what the authors study. While there are many recent theoretical works, both on feature learning, and on the behavior of transformers (see, e.g. [1-3]), not many prior works related to the topic of the paper are referenced. I believe this is yet another reason the manuscript does not motivate the main questions.

3. Some claims and statements seem unsubstantiated. I presume the authors do know why those statements hold, but the arguments are not communicated well. E.g., in line 295, the authors claim that they “*prove* attention can have a lower activation variance” through the analysis that follows. However, the analysis is on the Lipschitz constant, and the experiments do not seem to actually *prove* that (even in the sense of a “strong empirical evidence”, which still should not be referred to as a “proof”, especially in a paper that presents itself as a theoretical study).

4. The experiments do not show convincing evidence of the theory. While some match, some do not, and the experiments in the appendix seem inconsistent. Moreover, the inconsistencies are not properly discussed.


**Other weaknesses:**

a. The Preliminaries section is unnecessarily long. Batch normalization or MLP for instance, are basic concepts that could be stated in one line and the definitions could be moved to the appendix.

b. How did the authors use persistence homology or TDA in the paper at all? The abstract claims the use of TDA, and there is a subsection on persistence homology, but there seems to be no analysis based on TDA.

c. Some basic definitions and concepts are provided in the core theoretical results of the paper, where they do not belong. e.g., the definition of the Wasserstein distance, or Theorem 5, are too basic to be stated in detail in section 3. They should be either in the preliminaries or in the appendix.

d. While intuitive, the arguments in section 3.4 lack rigor. The authors later claim (line 485) that they “prove” that attention leads to a lower intrinsic dimensionality, while I do not see anything close to a proof in section 3.4 (and, again, experiments are not a proof, especially if the evidence is not thorough and strong).

e. There results in the appendix are not properly referenced, nor are they discussed in the main paper.

f. The authors claim that their analysis of the stability “has nothing to do with the model performance”) line 383, while there is often a tradeoff between expressivity (hence performance) and stability.

g. Arguments and explanations in lines 451-456 are not clear.

h. The discussion of Figure 2 seems unclear, with statements that do not clearly reference the evidence from the experiments.

i. This is a minor point, but I would not call Theorem 1 and 2 “theorem”, but rather a “proposition”, since they are direct results of the assumption that the input is a 0-mean Gaussian.



**References:**

[1] Von Oswald, J., Niklasson, E., Randazzo, E., Sacramento, J., Mordvintsev, A., Zhmoginov, A., & Vladymyrov, M. (2023, July). Transformers learn in-context by gradient descent. In International Conference on Machine Learning (pp. 35151-35174). PMLR.

[2] Abbe, E., Bengio, S., Boix-Adsera, E., Littwin, E., & Susskind, J. (2024). Transformers learn through gradual rank increase. Advances in Neural Information Processing Systems, 36.

[3] Radhakrishnan, A., Beaglehole, D., Pandit, P., & Belkin, M. (2024). Mechanism for feature learning in neural networks and backpropagation-free machine learning models. Science, 383(6690), 1461-1467.

**Questions:**

My main questions are implicit in the weaknesses mentioned. But two specific questions:

- Line 474: “We observe that while the distances in Attn become more variable after training, the distances in ViTs remain stable”. Where is this observed? (appendix? If so, why not reference or discuss?)

- How do you estimate the intrinsic dimension using kNN (line 476)? Perhaps this is standard, but I am not familiar with it, and the readers might not be either.

**Details Of Ethics Concerns:**

No ethics concerns.

---

> ### Author Response · Authors · 2024-11-20
>
> We thank the reviewer for their detailed and constructive feedback, which has helped us refine our manuscript. Below, we address the specific concerns raised:
>
> $\textbf{The key weakness}$
>
> After carefully considering the feedback from all reviewers, we decided to focus our study on the robustness of individual architectures, specifically comparing attention-based and convolutional models. In the revised version, we propose a robustness metric that enables both theoretical and empirical evaluation of robustness. This metric enables direct comparisons between the robustness of different model architectures. While the primary focus of the previous version was to compare attention and convolutional layers, in the revised version, the main goal has shifted to introducing and validating the proposed robustness metric. The comparison between attention and convolution serves as a validation of this metric.
>
> $\textbf{Major Weaknesses}$
>
> 1. In the revised version, we have clarified in the introduction why the proposed metric is needed and how it addresses gaps in existing methods. This discussion helps contextualize the relevance and importance of the work.
>
> 2. We provided literature review that is closely related to the topic in the revised version.
>
> 3. Thank you for pointing that out. The relationship between variance and the Wasserstein Lipschitzness indeed needs a theoretical analysis to connect them. However, in the revised version, we shifted focus entirely to Wasserstein Lipschitzness and no longer consider variance. The proposed robustness metric is now based solely on Wasserstein Lipschitzness, which is both theoretically and empirically validated.
>
> 4. We acknowledge the inconsistencies in the experimental results in the previous version, which may have stemmed from the unclear relationship between variance and Wasserstein Lipschitzness. After refocusing the paper, we provided experimental results that are consistent with the revised theoretical framework.
>
> $\textbf{Other weaknesses}$
>
> a. The background material has been shortened in the revised version.
>
> b. The Wasserstein distances reported are computed on persistence diagrams generated using persistence homology. We apologize for not stating this explicitly in the original version. The revised version includes a detailed explanation of how TDA is utilized.
>
> c. Theorem 5 from the previous version has been moved to the appendix as a lemma. The definition of the Wasserstein distance is retained in the main section, as it is essential for introducing Wasserstein Lipschitzness, which is central to our proposed metric.
>
> d. Intrinsic dimension analysis has been removed in the revised version, as it is less relevant to the paper’s new focus.
>
> e. The results in the appendix are now properly referenced and discussed in the main paper.
>
> f. We have included accuracies and losses for all models in the revised manuscript (see Table 2 and Figures 7 and 8).
>
> g. See lines 360-366 in the revised version for detailed explanation.
>
> h. Sorry for the unclearness. Figure 2 shows the Wasserstein distance of models at epoch 1, where models had not yet converged. Since these results do not represent model behavior appropriately, we removed the epoch-1 results and only keep the results that models converge in the revised version.
>
> i. Thank you for pointing that out.
>
> $\textbf{Questions}$
>
> $\bullet$ This observation was based on Figure 6 in the previous version. We apologize for not referencing it explicitly. In the revised version, we have ensured that all references to figures and results are clearly stated.
>
> $\bullet$ The (local) intrinsic dimension is computed by calculating the local dimensionality for each point by analyzing its k nearest neighbors and averaging the results to obtain a global estimate. However, as intrinsic dimension is no longer considered in the revised version, kNN methods are not used in this work.
>
> We hope that the revised manuscript addresses the reviewer’s concerns and clarifies the motivation, methodology, and findings of our work.

---

### Official Review · Reviewer_w8uD · 2024-11-03

**Soundness:** 3
**Presentation:** 2
**Contribution:** 2
**Rating:** 5
**Confidence:** 3

**Summary:**

This paper provides an analysis on the feature map variance between basic convnets and attention only networks. The author then expands this analysis to known architectures using the same building blocks such as the ResNet18 and the Vision Transformer. The author finds that their attention networks feature embeddings have a lower variance than those produced by convolutional nets, however this does not hold when expanding the attn network to a vision transformer architecture.

**Strengths:**

- The paper provides an interesting result, showing that attention mechanisms learn feature maps with lower variance than convolutional networks.
- The paper is mathematically correct from my understanding

**Weaknesses:**

- I am struggling to understand what is different between the attn -> vit models as well as the conv -> resnet. Is it the exclusion of residual connections and normalisation layers?
- There is no indication of the performance of the networks in accuracy, assuming they are at convergence I would expect the accuracies to be in the low to mid 90's for cifar-10, but the paper does not indicate whether the networks reached this performance or not. In my opinion, this is an important detail as otherwise we may be comparing underfitted networks which presumably do not have the correct feature space configuration for optimal accuracy.
- The author presents the statistics, but does little analysis into what this means. Do we want low variance or high variance in our feature space?
- (Minor) The figures 11-17 in the appendix are interesting visualisations of the feature space, but would be more informative if we zoomed in on the blobs of features to see if there is separation within the blobs, even if it is small.
- (Minor) To add onto the above point, visualisations of the class specific latent representations would be more interesting than the entire space to determine whether despite high or low variance of features, is our feature space semantic.

**Questions:**

I would like to state that currently I am very borderline on this paper as I struggle to find a clear takeaway other than statistics of the variance between different architectures feature spaces. I would appreciate if the author could address the points that I have made in the weaknesses section as well as clarifying what the takeaway of the paper is. I will always agree that our field needs more papers that provide methodologies to analyse the internals of our networks, but it is important that we also provide insight into what these analyses mean.

---

> ### Author Response · Authors · 2024-11-19
>
> We thank the reviewer for their detailed and constructive feedback, which has helped us refine our manuscript. Below, we address the specific concerns raised:
>
> $\textbf{A takeaway}$
>
> After carefully considering the feedback from all reviewers, we decided to focus our study on the robustness of individual architectures, specifically comparing attention-based and convolutional models. In the revised version, we propose a robustness metric that enables both theoretical and empirical evaluation of robustness. This metric enables direct comparisons between the robustness of different model architectures. While the primary focus of the previous version was to compare attention and convolutional layers, in the revised version, the main goal has shifted to introducing and validating the proposed robustness metric. The comparison between attention and convolution serves as a validation of this metric.
>
> $\textbf{Difference between attn -> vit as well as conv -> resnet}$
>
> Yes, the difference is the exclusion of residual connections and normalization layers. In the previous version, we concluded that ViTs have higher Wasserstein Lipschitz numbers than resnet18. However, we only considered resnet18 in the previous version, which uses basic blocks. In larger models such as resnet50 and resnet101, bottleneck blocks are typically used. In the revised version, we consider bottleneck blocks (for better generalization) and resnet50&101, finding that the empirical observations also align with the theoretical results for real-world models (see Section 4.2). It is important to note that the theoretical comparisons in this work are conducted under simplified settings (e.g., most entries follow $\mathcal{N}(0,\sigma^2)$ except for convolutional layers). While these settings provide valuable insights, they may not generalize to certain extreme scenarios. However, an important takeaway from the Lipschitzness bounds is that the Lipschitzness bounds show us the parameter dependencies of the smoothness of models (see l. 299-303).
>
> $\textbf{Experimental results}$
>
> We have added the training details in the revised version. The accuracies and losses of the models are also included in the revised version (see Table 2 and Figure 7 and 8).
>
> $\textbf{Low or high variance}$
>
> Both low and high variance in the feature space have advantages depending on the context: low variance indicates a more robust model, less sensitive to noise or perturbations, while high variance indicates greater expressiveness, capturing a wide range of input features.
>
> However, in the revised version, we no longer emphasize variance analysis. Instead, we focus on introducing and validating the proposed robustness metric (based on Wasserstein Lipschitness), which better aligns with the paper's main goal.
>
> $\textbf{Data visualization}$
> Given the shift in the paper’s focus, data visualizations are less central to validating the results, and therefore, we have removed them in the revised version. Nonetheless, we greatly appreciate the reviewer’s suggestions.
>
> We hope that these clarifications and the revisions made to the manuscript address the reviewer’s concerns.

---

### Official Review · Reviewer_cWa2 · 2024-11-03

**Soundness:** 2
**Presentation:** 2
**Contribution:** 2
**Rating:** 3
**Confidence:** 3

**Summary:**

The submission investigates the output variance and smoothness of non-residual (i.e., without skip connections) Attention and Convolutional layers. The authors demonstrate that attention is more compact and stable than convolution. Specifically, after presenting background on Transformers, Convolutions, and ResNets, the submission reviews recent results on the smoothness properties of attention in a mean-field framework. In Theorems 1 and 2, the input variance of the activations is derived, while Theorems 3 and 4 provide an upper bound on the Lipschitz constant of attention and convolution. Finally, experimental results on toy models (Conv and Attn), ResNet18, and a small ViT are presented, where activation variance, Wasserstein distances, and intrinsic dimensions are recorded across layers during training.

**Strengths:**

- The paper addresses an interesting question: understanding the theoretical differences in smoothness between attention-based and convolutional models, two widely used components of modern deep learning.
- The authors acknowledge that the theoretical findings do not transfer well to models with skip connections and normalization.
- The theoretical results appear rigorously proved.

**Weaknesses:**

In my view, this submission is not yet ready for publication at ICLR.

*In terms of writing:*
- The paper is challenging to follow. For instance, while the abstract and introduction repeatedly reference "feature learning," this term is not mentioned again in the rest of the paper. Another example is the mention of masked attention (l. 361), whereas no mask is used in ViTs. The motivation for using the $W_2$ metric, in my opinion, is not well explained (l. 307).
- The paper consists mostly of background material until page 5.
- The persistent homology part discussed in Section 2.5 is not referenced again in the rest of the submission.
- The related work section is very brief, with insufficient discussion of prior work. For instance, [2] also discusses the Lipschitz constant of self-attention.

*Regarding the theoretical contributions:*
- For attention, I am unsure about the novelty relative to previous work on the smoothness of attention, particularly Theorem 3.5 from Castin et al. (2024) and computations by Geshkovski et al. (2024).
- Theorem 5 should be replaced by a concrete result concerning deep ConvNets and Transformers rather than the actual theorem followed by a discussion (which I personally find unclear).

*Regarding the experimental contributions:*
- I am not convinced that the empirical results validate the tightness of the bounds in the theorems, or that they are sufficiently related to the rest of the paper.
- Additional experimental details in Section 4 are needed. How is activation variance computed? Over how many samples? Are these training or test samples?
- Generally, stacking attention layers without residual connections is very poor practice, as shown in [1] (which, in my opinion, should be discussed in the paper).

**Questions:**

- Could you clarify the differences between your proof technique and those of Geshkovski et al. and Castin et al.? What are the additional contributions?
- I would be very interested in seeing the accuracy of both models on CIFAR. Could the authors provide these numbers? I would especially expect the attention model without residual connections to perform poorly, as suggested by [1].
-The measure formulation of attention is also valid a finite number of tokens, correct?
- From what I understand, the empirical observations diverge from the theoretical results for real-world models (with residual connections and normalization) but align with the results for deep models without residual connections or normalization. Is this correct?

[1] Dong, Y., Cordonnier, J. B., & Loukas, A. (2021, July). *Attention is Not All You Need: Pure Attention Loses Rank Doubly Exponentially with Depth.* In International Conference on Machine Learning (pp. 2793-2803). PMLR.
[2] Kim, H., Papamakarios, G., & Mnih, A. (2021, July). *The Lipschitz Constant of Self-Attention.* In International Conference on Machine Learning (pp. 5562-5571). PMLR.

---

> ### Author Response · Authors · 2024-11-19
>
> We thank the reviewer for their detailed and constructive feedback, which has helped us refine our manuscript. Below, we address the specific concerns raised:
>
> $\textbf{Writing Problems}$
>
> After carefully considering the feedback from all reviewers, we decided to focus our study on the robustness of individual architectures, specifically comparing attention-based and convolutional models. In the revised version, we propose a robustness metric that enables both theoretical and empirical evaluation of robustness. This metric facilitates comparisons between different model architectures, offering insights into their robustness properties. While the primary focus of the previous version was to compare attention and convolutional layers, in the revised version, the main goal has shifted to introducing and validating the proposed robustness metric. The comparison between attention and convolution serves as a validation of this metric.
>
> In this revision, we have made the following key changes:
>
> 1. Mask attention: We no longer consider masked attention and instead focus on standard attention mechanisms. Sorry for the misunderstanding.
>
> 2. Metric choice: We now use the $W_1$ distance instead of $W_2$, using the fact that for two measures $\mu$ and $\nu$, $W_1(\mu,\nu) \leq W_2(\mu,\nu)$. This allows the computed Lip$^{W_1}$ to directly extend to Lip$^{W_2}$.
>
> 3. Background material: The background material has been shortened and integrated into relevant sections. For instance, the discussion of persistent homology has been combined with the introduction of the proposed method.
>
> $\textbf{Theoretical contributions}$
>
> $\bullet$ In previous works, the Lipschitz constant of attention includes exponential terms. In our revised version, we compute a tighter bound without exponential terms. See $\textbf{Proof of Theorem 1}$ for details.
>
> $\bullet$ We have provided concrete results for ResNets and Transformers in the revised version. See Section 4.2.
>
> $\textbf{Experimental contributions}$
>
> $\bullet$ We have streamlined the experimental results to align with the revised manuscript's focus. Specifically, we computed the change rate of the Wasserstein distance of persistence diagrams, which validates the proposed robustness metric. The results indicate that attention-based models are more robust than convolutional models, aligning with our theoretical predictions. Then we conduct robustness experiments using CIFAR-10C to verify the findings.
>
> $\bullet$ Activation variance is computed as described in l. 367-370 of the revised manuscript.
>
> $\bullet$ The practice of stacking attention layers without residual connections is to compare $\textit{TopoLip}$ (the metric proposed in the revised version, which stands for the maximum change rate of the Wasserstein distance). If including residual connection or layer normalization, it will be difficult to measure the Lipschitzness of attention (also the reason for the previous version).
>
> $\textbf{Questions}$
> 1. See the above answer.
>
> 2. See Table 2 and Figure 7 and 8 for accuracies and losses of models. Yes, the measure formulation of attention is valid for a finite number of tokens, but the number of tokens should be sufficiently large
>
> 3. Yes. However, we only considered resnet18 in the previous version, which uses basic blocks. In larger models such as resnet50 and resnet101, bottleneck blocks are typically used. In the revised version, we consider bottleneck blocks (for better generalization) and resnet50&101, finding that the empirical observations also align with the theoretical results for real-world models (see Section 4.2). It is important to note that the theoretical comparisons in this work are conducted under simplified settings (e.g., most entries follow $\mathcal{N}(0,\sigma^2)$ except for convolutional layers). While these settings provide valuable insights, they may not generalize to certain extreme scenarios. However, an important takeaway from the Lipschitzness bounds is that the Lipschitzness bounds show us the parameter dependencies of the smoothness of models (see l. 299-303).
>
> We hope that these clarifications and the revisions made to the manuscript address the reviewer’s concerns.

---

### Official Review · Reviewer_dL8z · 2024-11-03

**Soundness:** 2
**Presentation:** 3
**Contribution:** 1
**Rating:** 1
**Confidence:** 4

**Summary:**

The authors propose a mean-field regime study of attention and convolution, and argue the attention mechanism is more robust to variations in input data distributions, enabling more stable feature learning. They observe such conditions do not actually hold in ViTs, which are more aligned with ResNets in terms of behaviour. They demonstrate lower intrisinc dimensionality in feature learning of attention mechanisms wrt convolutional ones, however these characteristics do not persist in comparisons of ViTs and ResNets.

**Strengths:**

Strengths:
- Originality: I do not know other papers performing the same type of analyses. The authors develop or apply the theory in novel ways to draw some conclusions about the attention mechanism and convolutional layers. Therefore the work could be considered novel.
- Quality: the theoretical developments look sound with respect to the assumptions that are made ...
- Clarity: the paper is clearly written and easy to follow.
- Significance: unclear, not a strength point ...

**Weaknesses:**

I start the list of weaknessess with a complementary comment to the strenghts.

- Quality: ... however the experimental results are limited and undermine the usefulness of the theory.
- Significance: the significance or importance of the paper is not particularly clear. There does not seem to be any useful consequence of the theory developed. It is not clear what point the authors are trying to make because the lower variance and Wasserstein-lipschitz condition impact on possible applications (e.g. training stability and convergence, data efficiency, robustness, generalization, differential privacy etc.) are not mentioned or discussed (and if they are, it's more to state their irrelevance for real applications.  The authors should strongly motivate the utility of their study and how it produces insights that can lead to future useful developments. For instance, the authors could consider Differentially Private (DP) SGD training ,where batch normalization (BN) is not allowed and the lipschitzness (of the per-sample gradients in this case) has a strong impact on the training accuracy. If the authors could find a relationship between their wasserstein-lipschitzness and the ones of the per-sample gradients, it could have some practical impact on DP. Similarly, it would be interesting if the authors could find at least some toy practical applications in which their findings could show their possible impact.


- The CIFAR-10 experiments are conducted on extremely small models, it's unclear whether the findings generalise to both larger scale datasets or larger models. Furthermore, the training of transformers (and of CNNs too) is strongly regularised with augmentations and training tricks. It's not so clear whether the findings hold under such forms of regularization (see question about training details)
- Seveal works compare different aspects of transformers robustness and generalization. Many works have found that the claimed ability of attention mechanisms to focus on the whole of an input has little to no impact on the robustness of the learnt features, outlining that training tricks like pre-training and training procedures have larger impact than the inductive biases of the convolutional/attention mechanisms  [1,2,3]

[1] https://arxiv.org/abs/2207.11347
[2] https://arxiv.org/pdf/2310.16764
[3] https://arxiv.org/abs/2310.19909

**Questions:**

- How were the models trained? VITs, for how small, should not converge easily if trained from scratch on CIFAR-10. Did the authors use pre-trained models? Could you please provide full details about all the training details and procedure? Furthermore the absolute accuracies and losses of the models should be reported and compared (we are talking of learning dynamics, if the models do not show similar accuracies of lossess comparisons between models that have learnt completely different things, or haven't learnt anything at all don't make sense).

---

> ### Author Response · Authors · 2024-11-19
>
> We thank the reviewer for their detailed and constructive feedback, which has helped us refine our manuscript. Below, we address the specific concerns raised:
>
> $\textbf{Usefulness of the theory and the significance of the theory}$
>
> After carefully considering the feedback from all reviewers, we decided to focus our study on the robustness of individual architectures, specifically comparing attention-based and convolutional models. In the revised version, we propose a robustness metric that enables both theoretical and empirical evaluation of robustness. This metric facilitates comparisons between different model architectures, offering insights into their robustness properties.
>
> While the primary focus of the previous version was to compare attention and convolutional layers, in the revised version, the main goal has shifted to introducing and validating the proposed robustness metric. The comparison between attention and convolution serves as a validation of this metric. Additionally, we appreciate the reviewer’s suggestion to consider Differentially Private (DP) SGD training and related work. While this is beyond the scope of the current study, we acknowledge its importance and plan to explore such connections in future work.
>
> $\textbf{Toy practical applications}$
>
> We train models on CIFAR-10 and conduct robustness experiments on CIFAR-10C. The experimental aligns with the theoretical results, and it validates our method. We appreciate the reviewer’s suggestion regarding practical applications
>
> $\textbf{How were the models trained?}$
>
> All models are trained from scratch on CIFAR-10. In our settings, even the smallest ViT (6 heads, 384 embedding dimensions) converges.
>
> Training details: All models are trained on the training dataset of CIFAR-10 (Attns for 100 epochs, Convs for 200 epochs, ResNets for 100 epochs, and ViTs for 200 epochs). After convergence, we switch the models to evaluation mode to freeze their parameters. Then, we input the test dataset and collect the outputs from all layers and compute their persistence diagrams. We have added the training details in the revised version. The accuracies and losses of the models are also included in the revised version (see Table 2 and Figure 7 and 8).
>
> We hope this revised version addresses the reviewer’s concerns and clarifies the motivation, methodology, and significance of our work.

---

### Official Review · Reviewer_9Y3o · 2024-11-04

**Soundness:** 3
**Presentation:** 3
**Contribution:** 3
**Rating:** 6
**Confidence:** 2

**Summary:**

This paper presents a theoretical and empirical comparison of attention mechanisms and convolutional layers, focusing on their feature learning properties, including Lipschitz continuity, intrinsic dimensionality, and stability. It claims that attention mechanisms yield more stable and compact representations than convolutional layers and validates this through theoretical bounds and experiments on various architectures (Vision Transformers (ViTs) and ResNets).

**Strengths:**

* The paper provides a rigorous theoretical analysis of the feature learning characteristics of attention versus convolution

**Weaknesses:**

* Since attention and convolutional architectures can be combined in practice, it would have been useful to explore hybrid models or discuss scenarios where attention layers supplement convolutional layers, as is common in many architectures.

* Training on CIFAR-10 may not yield strong performance for ViTs, as they typically require large amounts of training data. Could this limitation have impacted the results?

**Questions:**

Look at the weaknesses.

---

> ### Author Response · Authors · 2024-11-19
>
> We thank the reviewer for their valuable feedback and comments.
>
> $\textbf{Exploration of hybrid models}$
>
> After carefully considering the feedback from all reviewers, we decided to focus our study on the robustness of individual architectures, specifically comparing attention-based and convolutional models. In the revised version, we propose a robustness metric and analyze the differences between these two architectures. Since the primary goal of our work is to demonstrate the robustness properties of attention and convolutional layers independently and validate the proposed metric, exploring hybrid models falls outside the scope of this study. Nonetheless, we appreciate the suggestion and recognize the potential of hybrid models for future investigations.
>
> $\textbf{Training on CIFAR-10 may not yield strong performance for ViTs}$
>
> In the revised manuscript, we have included the accuracy and loss of each model (Table 2 and Figure 7,8). ViTs achieve validation accuracies ranging from 77.8% to 87.0%, which may indicate a limitation due to the relatively small dataset size. However, since their performance is reasonably close to ResNets (around 91%), we believe that the ViTs are not significantly under-trained, and their performance suffices for our comparative analysis. Nonetheless, we acknowledge that larger datasets might yield even stronger results for ViTs.

---

### Note · Authors · 2024-11-21

I have read and agree with the venue's withdrawal policy on behalf of myself and my co-authors.